# Standardized and Quantitative ICG Perfusion Assessment: Feasibility and Reproducibility in a Multicentre Setting

**DOI:** 10.3390/life15121868

**Published:** 2025-12-05

**Authors:** Eline Feitsma, Hugo Schouw, Tim Hoffman, Sam van Dijk, Wido Heeman, Jasper Vonk, Floris Tange, Jan Koetje, Liesbeth Jansen, Abbey Schepers, Tessa van Ginhoven, Wendy Kelder, Gooitzen van Dam, Wiktor Szymanski, Milou Noltes, Schelto Kruijff

**Affiliations:** 1Department of Surgery, University Medical Centre Groningen, 9713 GZ Groningen, The Netherlands; e.a.feitsma@umcg.nl (E.F.); h.m.schouw@umcg.nl (H.S.); tim.hoffman@limis.com (T.H.); wido@spctr.tech (W.H.); l.jansen@umcg.nl (L.J.); w.kelder@mzh.nl (W.K.); m.e.noltes@umcg.nl (M.N.); 2Department of Nuclear Medicine and Molecular Imaging, University Medical Centre Groningen, 9713 GZ Groningen, The Netherlands; j.vonk@umcg.nl (J.V.); go@tracercro.com (G.v.D.); 3Department of Plastic Surgery, Karolinska Institutet, SE-171 77 Stockholm, Sweden; 4Department of Molecular Medicine and Surgery, Karolinska Institutet, SE-171 76 Stockholm, Sweden; 5LIMIS Development BV, 8934 AD Leeuwarden, The Netherlands; 6Faculty Campus Fryslân, University of Groningen, 8911 CE Leeuwarden, The Netherlands; 7Department of Surgery, Frisius MC, 8934 AD Leeuwarden, The Netherlands; j.h.koetje@umcg.nl; 8Department of Surgical Oncology and Gastrointestinal Surgery, Erasmus MC Cancer Institute, University Medical Centre Rotterdam, 3015 GD Rotterdam, The Netherlands; s.p.j.vandijk@erasmusmc.nl (S.v.D.); t.vanginhoven@erasmusmc.nl (T.v.G.); 9Department of Radiology, University Medical Centre Groningen, 9713 GZ Groningen, The Netherlands; 10Department of Surgery, Leiden University Medical Centre, 2333 ZA Leiden, The Netherlands; f.p.tange@lumc.nl (F.T.); a.schepers@lumc.nl (A.S.); 11Department of Surgery, Martini Hospital, 9728 NT Groningen, The Netherlands; 12Erasmus MC Academic Centre for Thyroid Disease, Department of Surgery, University Medical Centre Rotterdam, 3015 GD Rotterdam, The Netherlands; 13TRACER Europe BV, 9723 JJ Groningen, The Netherlands; 14Department of Medicinal Chemistry, Photopharmacology and Imaging, Groningen Research Institute of Pharmacy, University of Groningen, 9713 AV Groningen, The Netherlands; w.szymanski@umcg.nl

**Keywords:** indocyanine green, Near-Infrared Fluorescence, parathyroid perfusion, ICG Angiography, thyroidectomy, hypoparathyroidism, perfusion quantification, standardization

## Abstract

Indocyanine green near-infrared fluorescence (ICG-NIRF) imaging is widely used to assess tissue perfusion, yet its subjective interpretation limits correlation with postoperative parathyroid function. To address this, the Workflow model for ICG-angiography integrating Standardization and Quantification (WISQ) was developed. This exploratory prospective multicenter study evaluated the reproducibility of WISQ in adults undergoing total thyroidectomy at two Dutch university centres. Patients with contraindications to ICG or prior neck surgery were excluded. Intraoperative imaging used standardized camera settings with blood volume-adjusted ICG dosing, and perfusion curves were analyzed using predefined regions of interest. Eighty patients were included. Significant inter-centre variability was observed in maximum fluorescence intensity, inflow slope, and outflow slope (*n* = 30). At the lead centre, outflow was the most promising predictor of postoperative hypoparathyroidism (HPT) (median −0.33 [IQR −0.49–−0.15] a.f.u./s for HPT vs. −0.68 [−0.91–−0.41], *n* = 17, *p* = 0.08), although no parameter significantly predicted HPT. Repeated ICG injections consistently produced lower maximal intensities irrespective of injection rate, and reproducible curves were achieved only when ICG was freshly dissolved at 0.5 mg/mL instead of 2.5 mg/mL. These findings indicate that ICG concentration and injection technique influence perfusion kinetics and underscore the need to update WISQ with standardized injection dilution to improve its clinical utility.

## 1. Introduction

The assessment of tissue perfusion is of paramount importance in surgical settings, especially in procedures such as free-flap reconstructions, bowel anastomoses, or parathyroid-sparing thyroidectomy. Accurate evaluation of tissue perfusion is indispensable to prevent complications such as flap necrosis, anastomotic leakage, or hypoparathyroidism [1,2,3]. For instance, transient hypoparathyroidism occurs in approximately 19–38% of patients after thyroidectomy, while permanent hypoparathyroidism is reported in 15% of cases. Hypoparathyroidism reduces quality of life and increases the risk of mortality [4,5]. Similarly, anastomotic leakage occurs in 2–19% of colorectal surgeries, with mortality rates of 0.8–27% and adverse effects on long-term oncological outcomes [6]. Currently, no adequate techniques are available to guide intra-operative decision-making on optimal tissue perfusion. There is a pressing need for such a technique to minimize ischemia-related complications and enhance surgical outcomes.

Indocyanine green (ICG), a fluorescent dye binding to intravascular albumin, is increasingly used for intra-operative perfusion assessment [7]. However, the results of ICG near-infrared fluorescence (ICG-NIRF) imaging exhibit large variability, often relying solely on visual interpretation of fluorescent signal intensity. This results in subjective and irreproducible data, prohibiting the comparison of studies [8,9,10]. Moreover, factors such as dosage, injection time, camera distance to the surgical field, and camera specifications can cause variations in the fluorescent signal [11]. To address these issues, some studies have attempted to establish standardized ICG-NIRF imaging methods [12,13]. Time-dependent perfusion parameters and normalized fluorescence intensity metrics, where the maximum fluorescence intensity for each measurement is set to 100%, appear to provide superior predictive value for organ function compared to using absolute maximum fluorescence intensity values alone [12,14,15,16].

In 2021, Noltes et al. introduced the Workflow model for ICG-angiography integrating Standardization and Quantification (WISQ) to standardize and quantify parathyroid ICG-NIRF after total thyroidectomy, intending to expand this model for a broader surgical range [15,16]. Initially tested during thyroid surgery, this model was developed to predict postoperative hypoparathyroidism (HPT). HPT is a frequent complication linked to parathyroid vasculature damage during thyroid surgery, with an incidence of up to 15% [17,18]. Although WISQ was initially developed for measuring parathyroid perfusion, it can be applied to various tissues and organs. However, among all tissue types, the parathyroid gland provides an excellent model for refining WISQ since postoperative parathyroid function is dependent on adequate intact perfusion, and the parathyroid function can directly be monitored through calcium and parathyroid hormone (PTH) levels.

In this study, we looked at the feasibility of WISQ in a multicentre study, focusing on reproducibility and exploring multiple factors that influence the quantitative analysis of perfusion curves, such as injection speed and ICG concentration.

## 2. Materials and Methods

### 2.1. Study Design and Participants

This exploratory, prospective, and observational multicentre study was conducted in two Dutch university hospitals: the University Medical Centre Groningen (UMCG) and the Erasmus Medical Centre (EMC) in Rotterdam. Eligible subjects for inclusion were over 18 years of age and scheduled for total thyroidectomy for thyroid cancer, Graves’ disease, or goitre. Patients with known allergies to ICG or iodine, previous neck surgery, dialysis-dependent renal failure, a history of renal transplantation, and pregnant or breastfeeding women were excluded. Patients in whom the WISQ protocol was not correctly followed were identified based on abnormal curves. To further understand these abnormalities, a subgroup analysis was conducted. These patients were excluded from the final cohort analysis.

The study protocol received approval from the medical ethical review committee at the University Medical Centre Groningen (METc 201900307). It was conducted in accordance with the Declaration of Helsinki (adapted version Fortaleza, Brazil, 2013) and Good Clinical Practice guidelines. The study was registered in the clinicaltrials.gov registry (NCT06579430), but not prior to study initiation. However, the study protocol, including study procedures and endpoints, had been finalized and approved before data collection, and the study was conducted in accordance with this protocol. Written informed consent was obtained from all subjects prior to any study-related procedures.

### 2.2. Postoperative Hypoparathyroidism

Based on the literature, a 70–88% perioperative decrease in PTH is associated with the development of persistent HPT [18,19,20]. Here, we selected 80% perioperative decrease in PTH on the first postoperative day as threshold for postoperative HPT. For patients with incomplete pre- or postoperative PTH measurements, postoperative hypocalcaemia treatment (either oral or intravenous calcium suppletion) served as an indicator for clinical postoperative HPT.

### 2.3. Intra-Operative Imaging

Thyroidectomy was performed according to standard of care, employing capsular dissection to preserve the parathyroid glands and their vasculature as far as technically and oncologically possible, considering the patient’s disease and treatment goals.

Following completion of the thyroidectomy, haemostasis was secured. Before wound closure, ICG-NIRF imaging was performed using the Spectrum Platform 2.0™ (Quest medical imaging, Wieringerwerf, The Netherlands). Identical systems of this model, with the same specifications, were available in both centres, and imaging was performed according to the WISQ model [15,16]. According to this model, the camera lens was positioned at a fixed distance of 30 cm to the wound bed at a perpendicular angle, with the gain set at 22.5 decibels (dB) and an exposure time of 50 milliseconds (ms), as demonstrated by Noltes et al. in a video publication [16]. After that, ICG (Verdye^®^) was dissolved in sterile water at a concentration of 2.5 mg/mL, as recommended by the manufacturer (Diagnostic Green GmbH, Aschheim-Dornach, Germany). An amount of 1.5 mg of ICG per litre of circulating blood volume was then administered through manual bolus intravenous injection. Blood volume estimation was based on the patient’s height and weight. Injection rate and intraoperative haemodynamic parameters (e.g., mean arterial pressure and vasopressor use) were not standardized or systematically recorded and were therefore considered anticipated confounders in the interpretation of perfusion kinetics. A detailed, step-by-step overview of the WISQ imaging protocol is provided in Figure 1 (parts 1 and 2).

In case not all parathyroid glands could be captured in one field of view, imaging of both sides of the neck was conducted sequentially. For the second recording, the procedure was repeated with the same dose of ICG.

### 2.4. Multicentre Cohort

To evaluate the potential applicability of WISQ in a multicentre context, data from the UMCG and EMC were compared. For standardization purposes and comparability evaluation, the technical composite phantom developed by Gorpas et al. was used to compare the camera systems in both participating centres [21]. The fluorescence intensity of the phantom was measured and compared across the centres. This comparative analysis evaluates consistency and uniformity in imaging quality and performance among the various camera systems used across the different institutions.

### 2.5. Root Cause Analysis Cohort

An additional root cause analysis pilot aimed to assess the reproducibility of perfusion curves and the impact of injection speed and ICG concentration on parathyroid perfusion parameters in the lead centre (the UMCG). Inclusion and exclusion criteria aligned with the multicentre cohort. The procedure described in Figure 1 was repeated four to five minutes after the initial ICG administration, focusing on the same parathyroid gland in the same patient. The surgical procedure was paused in between the measurements. Both the effects on the parathyroid glands and on the skin were studied. Five patient groups were included (Table 1) to compare injection speed, camera systems, and ICG concentrations, while the remaining variables were left unchanged. Fast ICG administration was defined as an injection completed within approximately 2 s and slow administration was defined as an injection completed within 15–20 s. To eliminate any potential influence of the camera system on the observed effects, three patients were imaged using the Spy-Elite™ camera system (Stryker, Kalamazoo, MI, USA). This camera had also been used in the previous WISQ study. Data from these patients were analyzed using ImageJ Fiji™ (version 2.15.0). The last three patients received two rapidly injected doses of freshly dissolved ICG at a concentration of 0.5 mg/mL, in the same blood volume-dependent dose, as all previous patients. In these three patients, measurements were conducted on the skin.

### 2.6. Quantification Protocol

After surgery, the data was analyzed with the Quest Research tool v4.3 video software™. Regions of interest (ROIs) were drawn around the parathyroid glands and, when applicable, on the skin. During video acquisition, the surgeon indicated each gland with an instrument, providing the reference for ROI placement. ROIs were placed as a tight circular contour around the outlined parathyroid gland. An example is provided in Appendix A. All ROIs for both study sites were drawn manually by the two UMCG investigators (EF and HS) and cross-checked to minimize inter-operator variability. The fluorescence intensity in these ROIs was automatically tracked over time to generate perfusion curves (Figure 2). All automatically extracted parameters were subsequently reviewed and, when necessary, corrected manually if noise caused the software to misidentify specific points on the curve. A step-by-step overview of the analysis workflow is provided in Figure 1 (part 3).

To account for interpatient variability, perfusion curves from the skin ROI were used to calculate a parathyroid-to-skin ratio (PSR), assuming normal perfusion of the skin. The PSR was calculated by determining the ratio between the parameter of the parathyroid and the same parameter for the skin.

Normalized curves were created, with the F_max_ in each individual curve standardized to 100%. These normalized curves facilitated comparison by minimizing the influence of maximum intensity while accentuating the importance of the shape of the curve. Perfusion parameters were derived from both the absolute and the normalized perfusion curves (Figure 2).

Patients from the multicentre cohort in whom all four parathyroid glands could not be identified, either intraoperatively or through histopathological assessment of the resected specimen, were excluded from the study, as the inability to locate all glands precluded accurate correlation of perfusion parameters with parathyroid function. As in WISQ, we assumed that the presence of at least one well-perfused parathyroid gland is sufficient for adequate parathyroid function [22]. As such, only the least impacted gland (LIG) of each patient was included in the analysis. The LIG was determined for each different flow parameter. For both inflow and outflow, the gland with the steepest slope was analyzed. Additionally, the highest F_max_, the lowest normalized AUC for both inflow and outflow (indicating rapid inflow and outflow), and the lowest time to peak were selected (Figure 2).

In the root cause analysis, all visualized parathyroid glands were studied. Curves from the first and second ICG administrations were compared for each gland and the skin, and we also conducted a comparison of perfusion parameters.

### 2.7. Statistical Analysis

IBM SPSS Statistics™ version 28.0.1.0 (IBM Corp., Armonk, NY, USA, 2021) was used for statistical analysis. Values were presented as median plus interquartile range. Shapiro–Wilk tests were conducted to test for normality. Mann–Whitney U tests were conducted for non-normally distributed variables, while T-tests were employed for normally distributed variables. Wilcoxon’s rank tests of paired T-tests were conducted for paired measurements.

## 3. Results

### 3.1. Multicentre Cohort

A total of 65 patients were enrolled between November 2020 and September 2022, with 32 patients in the UMCG and 33 in the EMC. Patient characteristics are detailed in Appendix A. Inclusion and exclusion flowcharts for both the multicentre cohort and the root cause analysis cohort are included in Appendix A. In 36 patients (55%), four parathyroid glands were identified, either during surgery or by histopathological examination. In six patients, one or more of the parathyroid glands were not adequately visualized on the camera, preventing assessment of all four glands. Therefore, 30 patients were included in the analysis. Seventeen (57%) of them had surgery in the UMCG, and thirteen in the EMC (43%). In eight patients (27%), a second ICG administration was necessary to facilitate visualization of all parathyroid glands. The median time interval between two injections was 15 [4–19] minutes. Postoperative HPT developed in fourteen patients (47%). In one patient, no preoperative PTH was measured. A static example of parathyroid angiography can be found in Appendix A.

In the primary data analysis of all patients from both centres with four identified parathyroid glands, no significant differences were observed between the perfusion parameters (Figure 3) of patients with and without HPT. In addition, the normalized data for each patient and PSR for each parathyroid gland showed no significant differences between the HPT and non-HPT groups (Figure 3). The perfusion curves and corresponding perfusion parameters for all patients enrolled are accessible in Appendix A, respectively. When comparing perfusion curves from both centres, differences were observed in F_max_, inflow slope, and outflow slope (Figure 3). To assess the comparability of both camera systems, an agreement analysis was performed on the Gorpas et al. composite fluorescence phantom using a Q-Q plot and Bland–Altman methodology (Appendix A) [21]. The Bland–Altman analysis demonstrated a mean bias of 4.54 units (SD 7.16, 95% LoA −18.6–+9.49), indicating that the UMCG system reported slightly higher values compared to the EMC system in the phantom data. Although this difference results in relatively high percentual differences on low intensity values, there was no indication for proportional bias between the systems and overall, the systems appeared to yield similar results (Appendix A). Notably, analyzing only the patients from UMCG (*n* = 17), the centre where WISQ was developed, a stronger outflow slope was observed in patients without postoperative HPT (−0.68 a.f.u./s [−0.91–−0.41]) than in patients with postoperative HPT (−0.33 a.f.u./s [−0.49–−0.15]) (Figure 3); however, this was not a significant finding (*p* = 0.08).

### 3.2. Protocol Deviations

Patients from the multicentre cohort in whom the standardization protocol was inadvertently not strictly followed were not included in the overall or root cause analysis. Their data was evaluated separately for educational purposes only to improve understanding and interpretation of perfusion curves. Examples are illustrated in Figure 4. Abnormalities or deviations identified included ICG injection via a long intravenous line, which led to a delayed onset of fluorescence intensity and reduced F_max_, as well as an unintentionally shorter camera distance to the surgical field, resulting in F_max_ saturation. Additionally, the presence of an unexpected parathyroid adenoma as a comorbidity was identified, resulting in a notably steep inflow and outflow slope of the perfusion graph.

### 3.3. Root Cause Analysis Cohort

The additional root cause analysis pilot involved fifteen patients between June 2023 and May 2024 to assess WISQ’s reproducibility. The flowchart for inclusion can be found in Appendix A. In the first twelve cases, at least one parathyroid gland and the skin were visualized during total thyroidectomy or hemithyroidectomy. In the last three patients, only the skin was visualized to avoid analyzing poorly visible or damaged parathyroid glands. Figure 5a illustrates representative examples from the pilot cohort, and detailed statistical analyses are provided in Table 2. The interval between two injections was 5 [4.5–6] minutes.

A second injection of ICG, irrespective of injection speed, resulted in a lower F_max_ in twelve out of fifteen glands (52.6 a.f.u. [24.4–76.9] in the first injection versus 38.8 a.f.u. [21.5–49.5] in the second injection, *p* = 0.08). The inflow slope was lower in ten out of fifteen glands after the second injection (1.53 a.f.u./s [0.51–3.42] in the first injection versus 1.02 [0.59–2.88] in the second injection, *p* = 0.07), and the outflow slope was lower in eight out of nine glands (−0.36 [−1.04–0.34] versus −0.30 [−0.65–−0.28], *p* = 0.02, respectively). In six glands, 80% outflow was not reached in the first, the second, or both injections. Outflow could therefore not be compared in these glands. *p*-values were reported where applicable but should be interpreted with caution given the limited sample sizes. Injecting the solution more quickly (i.e., in approximately 2 s) resulted in a faster onset of fluorescence for both the first and second injections. For the slower injection, onset occurred at 32 s [31–33] (first injection) and 23 s [21–27] (second injection). In contrast, faster injection led to earlier onset times of 15 s [14–20] and 20 s [18–22], respectively. Injection speed did not affect the time to peak or the inflow slope.

To investigate whether the lower F_max_ value for the second injection and the delayed peak onset (T begin inflow) for slow injections were specific to the parathyroid gland, the same analysis was performed on the skin. Similar effects were observed when ROIs were drawn on the same place of the skin for the first and the second injections (Appendix A). Comparable results were found in group 4 (Table 1) using a different camera system, namely the Stryker Spy-Elite, thereby ruling out the camera system as a causal factor (Appendix A). When a lower concentration of 0.5 mg/mL was administered, with doses freshly dissolved before each administration, curves and parameters were similar for the first and second injections (Figure 5b). The parameters of these curves can be found in Appendix A.

## 4. Discussion

This exploratory study underscores the challenges in achieving reproducible ICG-NIRF imaging across different centres. Despite using similar camera systems and a standardized protocol, flow parameters were not comparable between centres. Although the EMC system yielded slightly higher Fmax values in patients, a direct comparison showed no relevant bias between cameras, making a systematic device effect unlikely. The observed variation is therefore more plausibly explained by centre-specific factors in ICG preparation and administration. Our additional root cause analysis identified several variables to be addressed towards reproducible ICG-NIRF, such as injection speed and possible ICG aggregation.

In the main centre, our findings suggest that poor outflow could be the best predictor for postoperative HPT. While Noltes et al. [15] described a correlation between compromised inflow and outflow and the occurrence of HPT, we could not consistently reproduce these results in a multicentre setting.

The additional root cause analysis pilot highlights the potential influence of various injection-related parameters, such as infusion speed and repeated ICG injections, on reproducibility. Injection speed, in particular, significantly affects inflow parameters compared to outflow parameters, although both exhibited considerable variability. These findings align with observations in radiology and angiography that correlating inflow parameters to outcomes becomes challenging without standardizing injection speed. Faster injection speed decreases time to peak and increases F_max_, while outflow remains more constant in other studies [23,24], although these specific outcomes are not seen in our small root cause analysis cohort.

A second ICG dose administered five minutes after the initial injection consistently yielded a lower F_max_. This phenomenon was also observed with a different camera system. Several biophysical and technological causes for this phenomenon were considered. A technical cause was ruled out as camera settings were fixed and no device-controlled parameters were present in the camera system. Second, quenching and albumin-binding saturation are unlikely explanations as the calculated plasma concentration after two injections remains low (3.15 µM or 2.5 µg/mL; Appendix A), far below the concentrations at which self-quenching has been reported (>80 µg/mL) [25].

A more plausible explanation involves concentration-dependent ICG aggregation between injections [26,27]. In this study, ICG powder was dissolved in water at 2.5 mg/mL, following supplier recommendations (2.5–5 mg/mL) [28,29]. However, several studies report a solubility limit of 1 mg/mL [30,31], and concentrations above 0.0077 mg/mL are already prone to aggregation [30,32]. While aggregation is less problematic for qualitative assessments, it hampers reliable quantification. At higher concentrations, rapid dimer and H-aggregate formation alters spectral and fluorescence properties [33,34,35,36]. This behaviour aligns with our observations: in patients receiving two freshly dissolved 0.5 mg/mL injections, fluorescence curves were comparable between injections, whereas consecutive injections from a 2.5 mg/mL solution consistently showed reduced intensities. Although the underlying mechanisms cannot be fully disentangled in this small cohort, the available evidence strongly supports aggregation at higher stock concentrations as a contributor to the observed variability. Together, these findings reinforce the recommendation to use ICG concentrations <1 mg/mL for quantitative angiography [37,38] and highlight the need for future in vitro and in vivo studies to further clarify the relative contributions of concentration and administration timing.

This study has several limitations. Firstly, half of the cohort had to be excluded from the analysis because fewer than four parathyroid glands were identified. This criterion was applied during analysis as the LIG can only be assigned when all four glands are accounted for and should be prospectively incorporated in future studies. Secondly, injection speed was not standardized, as optimal ICG administration rates remain insufficiently studied. Lastly, physiological factors such as blood pressure, heart rate, and medication were not considered. For example, norepinephrine has been shown to increase fluorescence intensity in bowel perfusion, but its broader impact on perfusion imaging is still unclear [39].

Despite its limitations, this study highlights the importance of standardizing factors such as ICG injection speed, concentration, and timing for an adequate perfusion assessment. Collaboration with radiologists, who are experienced in understanding these factors, is crucial [40]. Lessons from positron emitting tomography (PET) scan comparison across centres underline the need for centre-specific imaging calibration [41,42]. Intraoperative perfusion quantification may require the same approach to ensure comparative outcomes. Importantly, outflow emerged as a potential predictor of postoperative HPT, justifying further study in standardized multicentre settings.

## 5. Conclusions

Perfusion curves were not reproducible in this multicentre setting despite standardized and quantitative analysis, underscoring the challenge of multicentre application. The study’s multicentre design, combined with the analysis of a large number of perfusion curves and multiple variables affecting curve characteristics, highlights the complexity involved in translating ICG-NIRF imaging into consistent and clinically meaningful metrics. In this study, impaired parathyroid outflow, rather than inflow, emerged as the most indicative predictor of postoperative HPT, underlining its clinical potential. The observed inter- and intra-patient variability in ICG-NIRF perfusion parameters may be attributed to ICG molecule aggregation, affecting measurement consistency. To enhance reproducibility, future research should focus on refining the WISQ model by standardizing ICG preparation, using freshly dissolved ICG at concentrations below 1 mg/mL, and ensuring a consistent injection speed. With a more robust and validated protocol, a repeat multicentre study will be essential to confirm these findings and improve the clinical applicability of ICG-NIRF imaging in surgical practice.

## Figures and Tables

**Figure 1 life-15-01868-f001:**
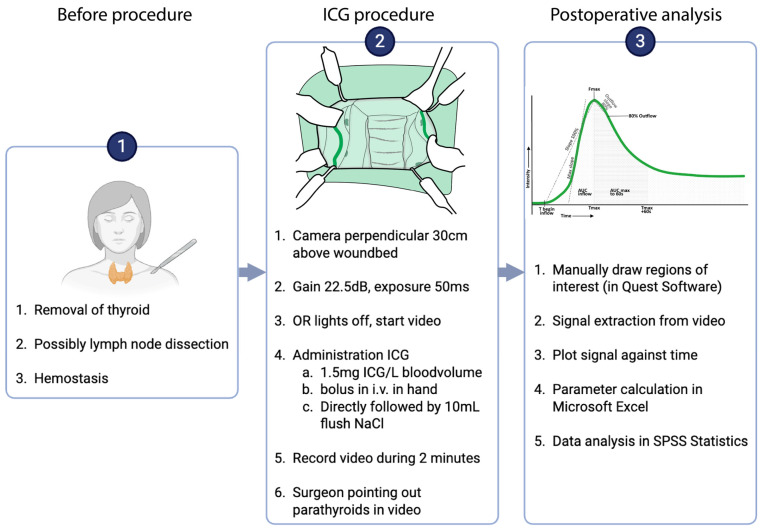
Intra-operative procedure and postoperative analysis procedure. In the root cause analysis pilot, step 2 was repeated on the same parathyroid glands.

**Figure 2 life-15-01868-f002:**
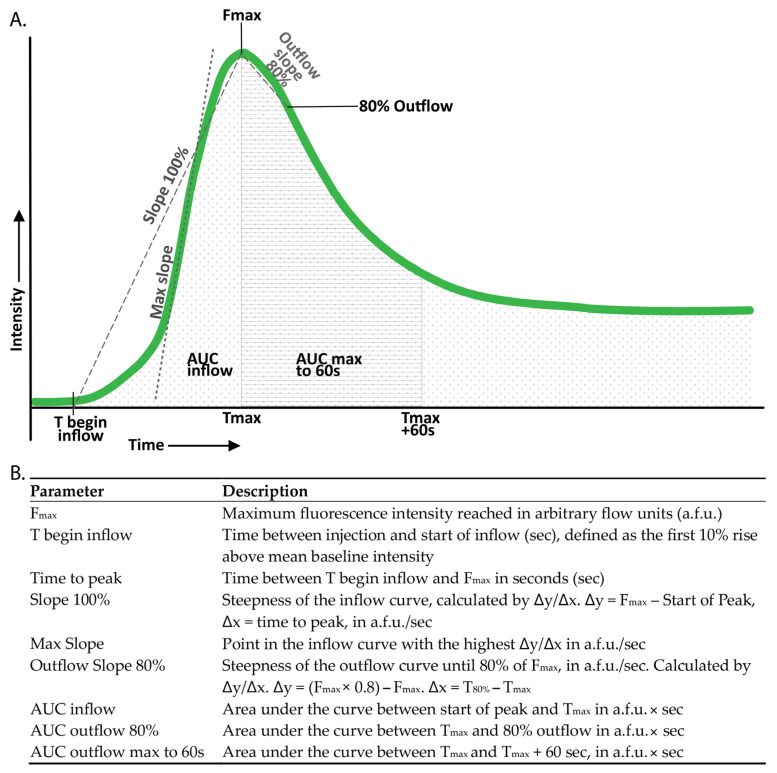
Definition of the studied perfusion parameters. (**A**) Perfusion curves for each region of interest (ROI) were plotted as fluorescence intensity against time. (**B**) Perfusion parameters were derived from these perfusion curves.

**Figure 3 life-15-01868-f003:**
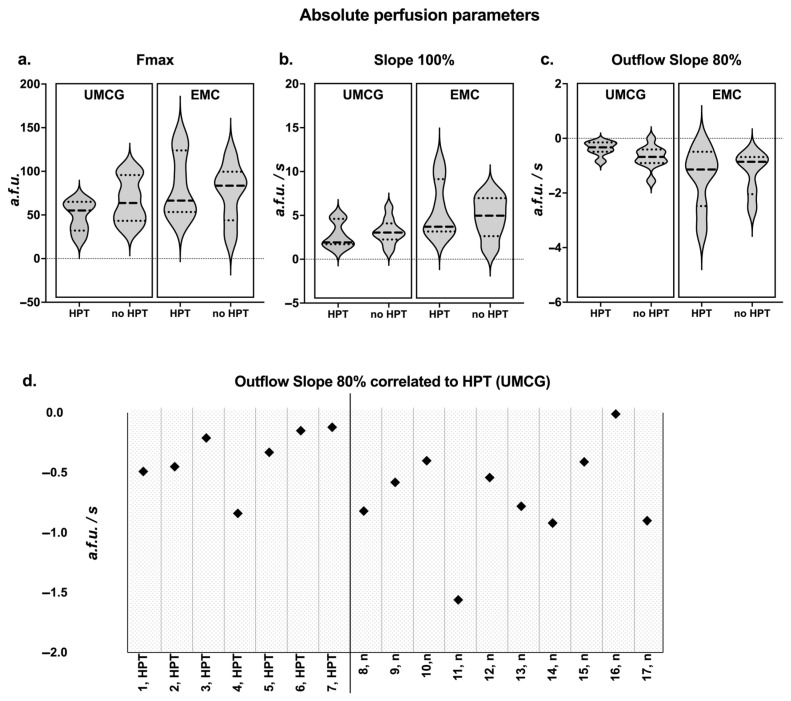
Quantification results of the multicentre cohort. (**a**–**c**) Absolute perfusion parameters in the multicentre cohort. Median, Q1, and Q3 are indicated on the violin plots. Data is presented for the two centres and for HPT versus non-HPT patients. All parameters can be found in Appendix A. (**d**) Outflow slope correlated to hypoparathyroidism (HPT (UMCG)): the outflow slope for patients from UMCG was plotted. Patients with HPT exhibit a poorer outflow compared to patients without HPT (n); however, this finding was not significant (*p* = 0.08).

**Figure 4 life-15-01868-f004:**
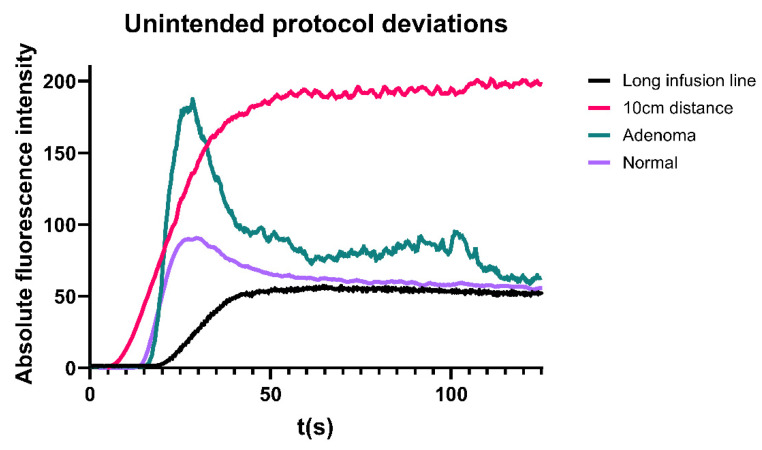
Perfusion curves in several patients in whom the protocol was not strictly adhered to.

**Figure 5 life-15-01868-f005:**
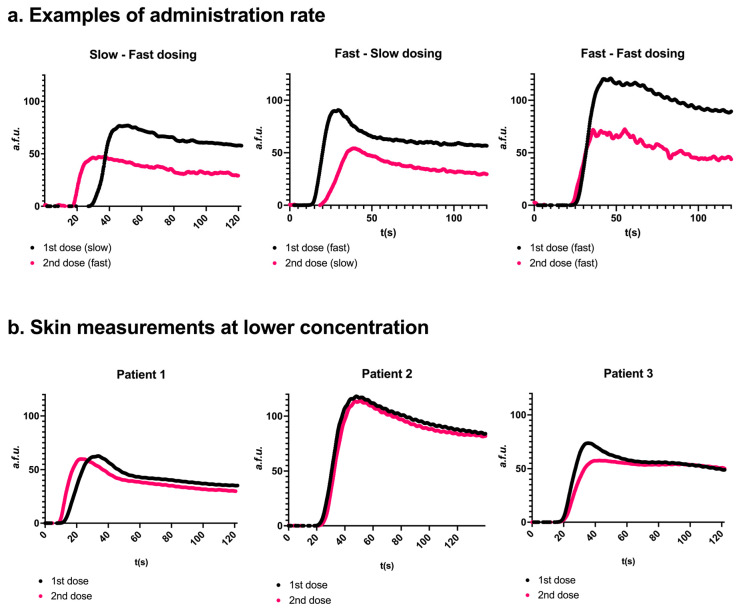
Effects of repeated injection and injection rate (panel (**a**)) or ICG concentration (panel (**b**)). (**a**) Three representative examples. Two consecutive measurements were conducted on the same parathyroid gland. A second dosing resulted in a lower F_max_. Slow injection resulted in a later peak onset compared to fast injection. Skin examples for ‘a’ can be found in Appendix A. (**b**) Measurements were obtained in 3 patients after administration of freshly dissolved ICG at a concentration of 0.5 mg/mL. The perfusion curves for the two consecutive administrations were more comparable to each other than a concentration of 2.5 mg/mL.

**Table 1 life-15-01868-t001:** Overview of the five groups in the root cause analysis cohort. Fast ICG administration was defined as injection within approximately 2 s and slow administration as injection within 15–20 s.

	Camera System	*n*	Injection Rate of 1st Administration	Injection Rate of 2nd Administration	Concentration
Group 1	Quest Platform 2.0	3	Slow	Fast	2.5 mg/mL
Group 2	Quest Platform 2.0	3	Fast	Slow	2.5 mg/mL
Group 3	Quest Platform 2.0	3	Fast	Fast	2.5 mg/mL
Group 4	Stryker Spy-Elite	3	Fast	Fast	2.5 mg/mL
Group 5	Quest Platform 2.0	3	Fast *	Fast *	0.5 mg/mL

* In the fifth group, rapid administration was performed; however, due to the larger volume, the rate exceeded the rates of groups 1 to 4, with the volume depending on the total injected dose. Injection duration within the same patient was similar for both injections.

**Table 2 life-15-01868-t002:** Median and interquartile ranges of parameters in the root cause analysis cohort.

			First			Second			
		*N*	Median	Q1	Q3	Median	Q1	Q3	*p*
T of Inflow Start (s)	Slow—Fast (group 1)	5	32	31	33	20	18	22	
	Fast—Slow (group 2)	4	15	14	20	23	21	27	
	Fast—Fast (group 3)	6	28	23	31	24	21	28	
	**1st vs. 2nd injection**	**15**	**28**	**18**	**32**	**22**	**20**	**25**	**0.19**
Fmax (a.f.u.)	Slow—Fast (group 1)	5	58.2	25.2	79.1	38.8	23.6	48.3	
	Fast—Slow (group 2)	4	54.1	40.9	82.9	38.8	27.7	50.5	
	Fast—Fast (group 3)	6	38.5	15.5	87.7	34.3	14.1	57.4	
	**1st vs. 2nd injection**	**15**	**52.6**	**24.4**	**76.9**	**38.8**	**21.5**	**49.5**	**0.08**
Time to Peak (s)	Slow—Fast (group 1)	5	37.4	17.4	93.5	17.0	12.9	103.9	
	Fast—Slow (group 2)	4	17.9	10.6	36.5	19.5	15.6	33.2	
	Fast—Fast (group 3)	6	45.2	18.0	60.0	31.8	18.4	38.8	
	**1st vs. 2nd injection**	**15**	**37.4**	**14.8**	**47.8**	**22.2**	**15.2**	**36.8**	**0.21**
Inflow Slope (a.f.u./s)	Slow—Fast (group 1)	5	1.53	0.40	2.69	1.28	0.28	3.40	
	Fast—Slow (group 2)	4	3.67	1.20	5.87	1.66	0.95	2.78	
	Fast—Fast (group 3)	6	0.85	0.39	2.82	0.87	0.49	2.37	
	**1st vs. 2nd injection**	**15**	**1.53**	**0.51**	**3.42**	**1.02**	**0.59**	**2.88**	**0.07**
Outflow Slope (a.f.u./s)	Slow—Fast (group 1)	2	−0.36			−0.30			
	Fast—Slow (group 2)	3	−1.54			−0.47			
	Fast—Fast (group 3)	4	−0.34	−0.49	−0.13	−0.29	−0.42	−0.15	
	**1st vs. 2nd injection**	**9**	**−0.36**	**−1.04**	**−0.34**	**−0.30**	**−0.65**	**−0.28**	**0.02 ***

* indicates statistical significance (*p* < 0.05)

## Data Availability

The original contributions presented in this study are included in the article/Appendix A. Further inquiries can be directed to the corresponding authors.

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
