# Peer review of "Standardized and Quantitative ICG Perfusion Assessment: Feasibility and Reproducibility in a Multicentre Setting"

_life, 2025, doi:10.3390/life15121868_

Round 1
Reviewer 1 Report
Comments and Suggestions for Authors
I enjoyed reading the paper "Standardized and quantitative ICG perfusion assessment: feasibility and reproducibility in a multicenter setting" by Eline Feitsma et al. This multicenter study evaluates the WISQ workflow for standardized and quantitative ICG-NIRF perfusion analysis in thyroidectomy and also includes an informative pilot study examining the effects of ICG administration rate and concentration. The topic is important and clinically relevant; the paper is well structured, the methods are described in detail, and the supporting materials are extensive. The figures and tables are easy to read, and the key results (in particular, the effect of administration protocol and the sustained decrease in Fmax with repeated high-concentration injections) are of practical value.
I have only a few comments that would be appreciated before publication.
Key comments
What is considered the "start of peak"? What is the threshold? Could noise distort the timing of this point, especially with slow administration? In general, with timings between Tbegin and Tmax of approximately 20 seconds, ICG administration over 10-15 seconds should certainly have an effect.
Minor/Editorial Edits
The text contains several "Error! Reference source not found." messages – this should be corrected.
Correct the orphan line (256) after Figure 4.
Median values and quartile lines are poorly discernible in violin plots (Figures 3a-c). We recommend overlaying transparent boxplots or enhancing the display of the median and IQR.
Typo in the list of abbreviations – Indocyanine Green
This work addresses a significant methodological issue, is well-executed, and, after addressing these issues, can be published with minor revisions.
Author Response
I enjoyed reading the paper "Standardized and quantitative ICG perfusion assessment: feasibility and reproducibility in a multicenter setting" by Eline Feitsma et al. This multicenter study evaluates the WISQ workflow for standardized and quantitative ICG-NIRF perfusion analysis in thyroidectomy and also includes an informative pilot study examining the effects of ICG administration rate and concentration. The topic is important and clinically relevant; the paper is well structured, the methods are described in detail, and the supporting materials are extensive. The figures and tables are easy to read, and the key results (in particular, the effect of administration protocol and the sustained decrease in Fmax with repeated high-concentration injections) are of practical value.
I have only a few comments that would be appreciated before publication.
Key comments
- What is considered the "start of peak"? What is the threshold? Could noise distort the timing of this point, especially with slow administration? In general, with timings between Tbegin and Tmax of approximately 20 seconds, ICG administration over 10-15 seconds should certainly have an effect.
Response:
We thank the reviewer for the positive assessment of our work and for this helpful comment. In our analysis, the start of peak (Tbegin inflow) was defined as the moment at which the fluorescence signal displayed a 10% increase relative to the mean baseline intensity. This threshold is automatically calculated by the QUEST software and was subsequently verified by the study team. Baseline noise in our dataset was minimal and substantially lower than the fluctuations occurring once ICG entered the circulation; therefore, a 10% rise above baseline did consistently represent the true onset of inflow.
We evaluated whether injection speed could introduce variability in this point. Faster injections consistently resulted in an earlier onset of fluorescence, as shown in Supplementary Information I. In contrast, we did not observe a consistent effect of injection speed on time to peak or inflow slope within this small cohort. Therefore, while injection rate influences the absolute onset of inflow, we found no evidence that it affected the determination of the start of peak as defined by our 10% threshold. However, these observations are based on a cohort of only 15 patients, and the sample size does not allow firm conclusions to be drawn.
To improve clarity for readers, we have made the following alterings to our manuscript:
-
-
- In figure 2 on page 6 the definition of T begin inflow was updated: “Time between injection and start of inflow (sec), defined as the first 10% rise above mean baseline intensity”
- In the methods section we clarified that all automated parameters were manually reviewed and adjusted when required: “All automatically extracted parameters were subsequently reviewed and, when necessary, corrected manually if noise caused the software to misidentify specific points on the curve.” Page 5, line 186 - 188
-
Minor/Editorial Edits
The text contains several "Error! Reference source not found." messages – this should be corrected.
Response: We thank the reviewer for pointing this out. During submission, some of our automatic cross-references to tables and figures were lost, resulting in “Error! Reference source not found.” messages in the text. We have now carefully corrected all cross-references throughout the manuscript.
Correct the orphan line (256) after Figure 4.
Response: In the revised version of the manuscript, no orphan line appears after Figure 4, and we trust that this resolves the issue.
Median values and quartile lines are poorly discernible in violin plots (Figures 3a-c). We recommend overlaying transparent boxplots or enhancing the display of the median and IQR.
Response: We have revised the figure and believe it is now clearer and easier to read.
Typo in the list of abbreviations – Indocyanine Green
Response: We have changed the typos in the list of abbreviations
This work addresses a significant methodological issue, is well-executed, and, after addressing these issues, can be published with minor revisions.
Reviewer 2 Report
Comments and Suggestions for Authors
This study is relevant and contributes to the standardization of ICG-NIRF imaging for parathyroid gland perfusion during thyroidectomy by addressing the key issue of subjective interpretation. The WISQ model shows promise, but improvements in clarity and data presentation are needed to improve its efficiency.
- The WISQ model plays a key role, but the details of its implementation are poorly described. For example, how were regions of interest (ROIs) defined and drawn? Were they automated or manual? Differences in ROI placement between operators may explain some differences between centers (e.g., peak fluorescence intensity, inflow/outflow slopes). Appendix D ("All patients curves.docx") presents curves but lacks ROI examples. Expand the Methods section to include a step-by-step description of the WISQ protocol, including ROI criteria (e.g., size, freedom from artifacts). Add an additional figure illustrating ROI placement in sample images.
- The results obtained with repeated injections (at lower intensities) and dilutions (0.5 mg/mL fresh solution versus 2.5 mg/mL) are informative and are supported by the calculations provided (e.g., the albumin to ICG ratio is ~215:1, which rules out saturation). However, the primary cohort (n=15) is small, and factors such as injection speed (fast/slow) are poorly understood. Please support the discussion with explanations (e.g., cite literature data on ICG aggregation or concentration-dependent fluorescence quenching at higher concentrations).
- The text should be corrected for errors: Error! Reference source not found.
Author Response
Response to reviewer
We sincerely thank the reviewer for the extensive and thoughtful feedback on our manuscript. We greatly appreciate the care and expertise reflected in these comments.
Reviewers comments:
The WISQ model plays a key role, but the details of its implementation are poorly described. For example, how were regions of interest (ROIs) defined and drawn? Were they automated or manual? Differences in ROI placement between operators may explain some differences between centers (e.g., peak fluorescence intensity, inflow/outflow slopes). Appendix D ("All patients curves.docx") presents curves but lacks ROI examples. Expand the Methods section to include a step-by-step description of the WISQ protocol, including ROI criteria (e.g., size, freedom from artifacts). Add an additional figure illustrating ROI placement in sample images.
Response: We thank the reviewer for this helpful comment and for prompting further clarification of the WISQ implementation. We have made the following changes:
- ROI definition and placement: We have expanded the Methods section to clarify how ROIs were defined and drawn. Specifically, we now state that ROIs were drawn manually by the two UMCG investigators (EF and HS) for all patients from both centres, using the surgeon’s intraoperative indication of each parathyroid gland during video acquisition as reference. ROIs were placed as a tight circular contour around the outlined parathyroid gland and were cross-checked between investigators to minimise inter-operator variability. This also explicitly clarifies that ROI placement was manual rather than automated. “During video acquisition, the surgeon indicated each gland with an instrument, providing the reference for ROI placement. ROIs were placed as a tight circular contour around the outlined parathyroid gland. An example is provided in Supplementary Information A. All ROIs for both study sites were drawn manually by the two UMCG investigators (EF and HS) and cross-checked to minimise inter-operator variability.” Page 5, line 180 - 185
- Step-by-step description of the WISQ protocol: In the Methods, we now explicitly refer to a step-by-step description of the WISQ protocol for both the procedural part (camera set-up, ICG preparation, dosing and injection, image acquisition) and the analysis workflow (ROI placement, generation of perfusion curves, parameter extraction and PSR calculation), as illustrated in Figure 1.1–1.3.
- “A detailed, step-by-step overview of the WISQ imaging protocol is provided in Figure 1.1 and 1.2.” Page 3, line 137-138
- “A step-by-step overview of the analysis workflow is provided in Figure 1.3.” Page 5, line 188 - 189
- As requested, we have added an additional figure in Supplementary Information A showing example images with ROI placement around the parathyroid gland, illustrating how ROIs were positioned in practice.
The results obtained with repeated injections (at lower intensities) and dilutions (0.5 mg/mL fresh solution versus 2.5 mg/mL) are informative and are supported by the calculations provided (e.g., the albumin to ICG ratio is ~215:1, which rules out saturation). However, the primary cohort (n=15) is small, and factors such as injection speed (fast/slow) are poorly understood. Please support the discussion with explanations (e.g., cite literature data on ICG aggregation or concentration-dependent fluorescence quenching at higher concentrations).
- Response: We agree that the primary cohort is small and that certain variables, such as injection speed, cannot yet be fully disentangled. We have revised the relevant part of the Discussion to more clearly explain the biophysical mechanisms that may account for the consistently lower Fmax after a second injection.
Although this section already included extensive supporting literature, we now more explicitly clarify why albumin-binding saturation and self-quenching are unlikely, noting that the calculated plasma concentration after two injections (3.15 µM; 2.5 µg/mL) remains far below reported quenching thresholds (>80 µg/mL). We also strengthened the explanation regarding concentration-dependent aggregation, highlighting the discrepancy between the recommended reconstitution concentration (2.5–5 mg/mL) and the reported solubility limit (~1 mg/mL), and linking these concepts more directly to our observations. The revised text underscores that reproducible curves were observed with two fresh 0.5 mg/mL injections, whereas repeated injections from a 2.5 mg/mL solution consistently yielded lower intensities, supporting aggregation at higher stock concentrations as the most plausible explanation.
While our cohort is indeed small, we believe that this mechanistic perspective is highly relevant for the broader use of ICG, particularly when quantitative interpretation is intended. To ensure that this section is accurately framed, we also discussed the paragraph with our co-author Prof. Wiktor Szymański, whose expertise helped finding the right literature and wording.
We hope that these clarifications adequately address the reviewer’s concern and improve the clarity of the discussion. (Page 13, Line 335-354)
The text should be corrected for errors: Error! Reference source not found.
- Response: We thank the reviewer for pointing this out. During submission, some of our automatic cross-references to tables and figures were lost, resulting in “Error! Reference source not found.” messages in the text. We have now carefully corrected all cross-references throughout the manuscript.
Reviewer 3 Report
Comments and Suggestions for Authors
I read the manuscript by Feitsma et al. with great interest. It addresses an important and timely topic, i.e., the intraoperative use of ICG fluorescence imaging for quantifying parathyroid perfusion during thyroidectomy and the feasibility and reproducibility of a standardized workflow in a multicenter setting. The study combines a prospective multicentre cohort with a focused root-cause analysis on ICG concentration and injection technique, and overall demonstrates strong technical methodology, detailed imaging standardization, and careful curve-based quantification. As per my standard practice, I evaluated the manuscript against the STROBE principles for observational studies.
Major comments
1) Post-hoc Exclusions: Only 30/65 patients were analysed due to exclusion of cases with <4 identified parathyroid glands. This criterion is not pre-specified and introduces potential selection bias. Please justify the analytic decision and provide baseline comparisons of included vs. excluded patients.
2) Outcome Heterogeneity: Hypoparathyroidism is defined using two distinct criteria (PTH-based vs. clinical). It is advisable to recommend a sensitivity analysis restricted to cases with complete PTH data to assess robustness.
3) Lack of a priori primary endpoint and power analysis: Multiple perfusion parameters were tested without specifying a primary variable. Additionally, no sample size or power justification has been provided. The analysis should be explicitly labelled as exploratory, and conclusions moderated accordingly.
4) Potential confounders not addressed: The injection rate of ICG and intraoperative haemodynamics (MAP, vasopressors) were not captured despite known influence on bolus geometry. These should be acknowledged in the Methods as anticipated confounders, not only in Discussion.
5) Retrospective registration: The registration occurred after study initiation. Please clarify the timing and confirm that endpoints and analyses were fixed prior to data examination.
Minor comments
-- The Abstract should report the final analysed sample (n=30) and explicitly note that no perfusion parameter significantly predicted HPT.
-- Please include p-values and effect sizes in the text and figures, regardless of the small sample size.
-- For completeness, the authors may also find helpful a relevant publication from our group [Surg Endosc 2024 Feb;38(2):511-528. doi: 10.1007/s00464-023-10546-4], which evaluates methodological heterogeneity and quality across surgical ICG applications. This may help contextualize the challenges the authors encountered. This is entirely optional.
Thank you once again for giving me the opportunity to review this interesting article. I look forward to receiving feedback from the authors.
Author Response
Response to reviewer
We thank the reviewer for the very kind and encouraging comments on our work. We greatly appreciate the recognition of the relevance of this topic, as well as the acknowledgement of the methodological rigor, imaging standardization, and quantitative approach used in our study.
Major comments
1) Post-hoc Exclusions: Only 30/65 patients were analysed due to exclusion of cases with <4 identified parathyroid glands. This criterion is not pre-specified and introduces potential selection bias. Please justify the analytic decision and provide baseline comparisons of included vs. excluded patients.
- Response: We thank the reviewer for the feedback. The exclusion of patients with fewer than four identifiable parathyroid glands was indeed not pre-specified; this decision was made during data analysis because reliable assignment of the least-impacted gland requires complete identification of all four glands. When one or more glands cannot be accounted for, the presence of a well-perfused remaining gland cannot be excluded, making perfusion–function correlation unreliable. To address the potential risk of selection bias, we added a comparison of baseline characteristics between included and excluded patients in Supplementary Information B, which shows no meaningful differences between groups. We now also note in the Discussion that this criterion should be predefined as an exclusion criterion in future studies. “This criterion was applied during analysis as the LIG can only be assigned when all four glands are accounted for, and should be prospectively incorporated in future studies.” Page 13, line 356 - 358
2) Outcome Heterogeneity: Hypoparathyroidism is defined using two distinct criteria (PTH-based vs. clinical). It is advisable to recommend a sensitivity analysis restricted to cases with complete PTH data to assess robustness.
- Response: It is correct that hypoparathyroidism was defined using two criteria (PTH-based vs. clinical), but among the analysed cohort this heterogeneity concerned only one patient, who had clinically defined hypoparathyroidism. As the results were very diverse and exclusion of this single case did not alter any of the findings, a separate sensitivity analysis would not change the conclusions. To improve clarity, we have now reported in Supplementary Information B how many patients met each definition. Although complete PTH measurements at fixed postoperative time points would technically allow more uniform outcome classification, this is not always feasible in routine clinical workflows.
3) Lack of a priori primary endpoint and power analysis: Multiple perfusion parameters were tested without specifying a primary variable. Additionally, no sample size or power justification has been provided. The analysis should be explicitly labelled as exploratory, and conclusions moderated accordingly.
- Response: We agree that no a priori primary endpoint or power calculation was defined. As stated in the Methods, this study was designed as an exploratory investigation aimed primarily at understanding sources of variability in ICG-derived perfusion curves and optimising the use of ICG angiography, rather than identifying a single predictive parameter for hypoparathyroidism. The analysis of multiple perfusion parameters therefore reflects this exploratory intent. To clarify this, we have now explicitly labelled the study as exploratory throughout the manuscript and have moderated the conclusions accordingly.
- “This exploratory prospective multicenter study evaluated the reproducibility of WISQ in adults undergoing total thyroidectomy at two Dutch university centers.” Page 1, abstract, line 31
- “In this study, we looked at the feasibility of WISQ in a multicentre study, focusing on reproducibility and exploring multiple factors that influence the quantitative analysis of perfusion curves, such as injection speed and ICG concentration.” Page 2, Introduction, line 86-88
- “This exploratory study underscores the challenges in achieving reproducible ICG-NIRF imaging across different centres.” Page 12, discussion, line 309
4) Potential confounders not addressed: The injection rate of ICG and intraoperative haemodynamics (MAP, vasopressors) were not captured despite known influence on bolus geometry. These should be acknowledged in the Methods as anticipated confounders, not only in Discussion.
- Response: Injection rate and intraoperative haemodynamics were indeed not systematically captured, and we agree that these factors may influence bolus geometry and fluorescence kinetics. We have now explicitly acknowledged these variables in the Methods as anticipated confounders.
- “Injection rate and intraoperative haemodynamic parameters (e.g., mean arterial pressure, vasopressor use) were not standardised or systematically recorded and were therefore considered anticipated confounders in the interpretation of perfusion kinetics.” Page 3, line 134-137
5) Retrospective registration: The registration occurred after study initiation. Please clarify the timing and confirm that endpoints and analyses were fixed prior to data examination.
- Response: The study was conducted as a non-WMO (non–Medical Research Involving Human Subjects Act) study, and was inadvertently not registered prior to enrolment. The registration was completed before manuscript preparation. All study procedures, endpoints, and planned analyses had been defined in the protocol approved by the ethics committee prior to data collection and prior to any data examination, and the study was conducted accordingly. We have updated the passage regarding the registration.
- “The study was registered in the clinicaltrials.gov registry (NCT06579430), but not prior to study initiation. However, the study protocol, including study procedures and endpoints, had been finalised and approved before data collection, and the study was conducted in accordance with this protocol.” Page 3, line 106-109
Minor comments
- The Abstract should report the final analysed sample (n=30) and explicitly note that no perfusion parameter significantly predicted HPT.
- Response: This has been addressed
- Please include p-values and effect sizes in the text and figures, regardless of the small sample size.
- Response: We added the p values in the text and figures throughout the article
- For completeness, the authors may also find helpful a relevant publication from our group [Surg Endosc 2024 Feb;38(2):511-528. doi: 10.1007/s00464-023-10546-4], which evaluates methodological heterogeneity and quality across surgical ICG applications. This may help contextualize the challenges the authors encountered. This is entirely optional.
- Response: We have added this helpful reference in our introduction as we think it perfectly fits the message of our article
Thank you once again for giving me the opportunity to review this interesting article. I look forward to receiving feedback from the authors.